# Drug repositioning of polaprezinc for bone fracture healing

Eun Ae Ko[1,4], Yoo Jung Park[1,4], Dong Suk Yoon [1,4], Kyoung-Mi Lee[1,2], Jihyun Kim[1], Sujin Jung[1,2], Jin Woo Lee [1,2,3] & Kwang Hwan Park [1✉]

Fractures and related complications are a common challenge in the field of skeletal tissue engineering. Vitamin D and calcium are the only broadly available medications for fracture healing, while zinc has been recognized as a nutritional supplement for healthy bones. Here, we aimed to use polaprezinc, an anti-ulcer drug and a chelate form of zinc and L-carnosine, as a supplement for fracture healing. Polaprezinc induced upregulation of osteogenesis-related genes and enhanced the osteogenic potential of human bone marrow-derived mesenchymal stem cells and osteoclast differentiation potential of mouse bone marrow-derived monocytes. In mouse experimental models with bone fractures, oral administration of polaprezinc accelerated fracture healing and maintained a high number of both osteoblasts and osteo-clasts in the fracture areas. Collectively, polaprezinc promotes the fracture healing process efficiently by enhancing the activity of both osteoblasts and osteoclasts. Therefore, we suggest that drug repositioning of polaprezinc would be helpful for patients with fractures.

[1] Department of Orthopaedic Surgery, Yonsei University College of Medicine, Seoul, South Korea. [2] Severance Biomedical Science Institute, Yonsei University College of Medicine, Seoul, South Korea. [3] Brain Korea 21 PLUS Project for Medical Science, Yonsei University College of Medicine, Seoul, South Korea. [4]These authors contributed equally: Eun Ae Ko, Yoo Jung Park, Dong Suk Yoon. ✉email: khpark@yuhs.ac

Zinc is an essential element that functions as a structural component in the human body, playing important roles in several biological processes, including DNA to protein synthesis[1]. Zinc is relatively abundant in skeletal tissues, such as bone, cartilage, and teeth[2], and zinc deficiency is known to delay bone growth and increase skin fragility[3]. Interestingly, research has shown that zinc supplementation has positive effects on bone health, including maintaining bone mineral density (BMD) and accelerating fracture healing[4,5]. In addition, zinc has been shown to stimulate the proliferation and differentiation of osteoblasts by stimulating collagen synthesis to improve bone formation[6,7]: along with vitamin C and copper, zinc is regarded as an important cofactor for collagen production[8], wherein zinc activates proteins responsible for collagen synthesis and zinc deficiency reduces that the amount of collagen produced[9]. Previous studies have demonstrated that zinc acts a signaling molecule in various intracellular signaling pathways[10,11], and our research has indicated that zinc is involved in the calcium-calcineurin-NFATc1 signaling pathway to inhibit osteoclast differentiation in mouse bone marrow-derived monocytes (mBMMs), as well as the AMP-PKA-CREB signaling pathway to promote osteoblast differentiation in human bone marrow-derived mesenchymal stem cells (hBMSCs)[12,13]. We suspect that zinc may be involved in more intracellular signaling pathways. Therefore, in order to properly apply zinc to bone-related diseases, links between zinc and various signaling pathways need to be established.

Polaprezinc is an oral bioavailable chelate consisting of zinc and L-carnosine with potential gastrointestinal protection, and antioxidant, anti-ulcer, and anti-inflammatory properties[14–17]. In clinical trials, polaprezinc was applied as a replacement therapy for zinc deficiency[18]. As zinc deficiency interferes with fracture healing[19], polaprezinc administration may also be considered a potential therapeutic agent to improve bone regeneration or increase the effectiveness of osteoporosis treatment. The combination or chelation of zinc and carnosine, which results in polaprezinc, has superior health benefits compared to individual treatment as carnosine enhances the absorption of zinc because of its solubility and that it possibly delivers zinc to the tissues in a delayed and extended release manner[20,21]. The chemical reaction of polaprezinc has been shown to release zinc during intestinal absorption in vivo[22]. This means that polarprezinc dissociate into L-carnosine and zinc during intestinal absorption[23]. Since polaprezinc is dissolvable in acid and appears to protect mucosal lesions, it can be thought that polaprezinc dissociates into two compounds during intestinal absorption[24]. Hydrochloric acid (HCL) solution is mainly used to dissolve polaprezinc in in vitro experiments, which means that the in vitro delivery method of polaprezinc by acidic solution makes it possible to dissociate polaprezinc into L-carnosine and zinc to mimic in vivo conditions. Therefore, in vitro experiments with polaprezinc may be suitable to study changes in vivo following oral administration. However, to date, there have been no reports on whether the oral administration of polaprezinc has a positive effect on fracture healing, as well as modulating the molecular mechanisms for osteoblast and osteoclast homeostasis in vitro. Drug repositioning is a way to investigate exiting drugs or molecules for other therapeutic purposes and can be a strategy for identifying new uses for approved drugs that are beyond the scope of their intended use[25]. Utilizing drug repositioning can minimize expensive and time-consuming processes for identifying new indications for drugs that have already been approved for patient use. In this study, we employed polaprezinc as a strategy for drug repositioning. Here, we showed that polaprezinc enhanced the differentiation potential of hBMSCs and mBMMs into osteoblasts and osteoclasts, respectively. Polaprezinc treatment upregulated the protein level of yes-associate protein (YAP), which is a

positive regulator of osteoblast and osteoclast differentiation, in precursors of osteoblasts and osteoclasts, respectively. In the study of mice with bone fractures, the oral administration of polaprezinc enhanced fracture healing. Our results indicate that polaprezinc is a potential supplement for fracture healing.

## Results

### Polaprezinc promotes osteoblast differentiation in hBMSCs.
To investigate whether polaprezinc could be a candidate small molecule for drug repurposing to bone-related diseases, we used hBMSCs and mBMMs to determine whether polaprezinc acts as a positive or negative modulator during osteoblast and osteoclast differentiation (Fig. 1a). Our previous study showed that zinc sulfate promotes osteoblast differentiation via the PKA-CREB signaling pathway[13]. We wanted to test whether polaprezinc could also act as a positive inducer of the osteogenic differentiation of hBMSCs. Before determining the effect of polaprezinc on osteoblast differentiation, we first tested the cytotoxicity of polaprezinc in hBMSCs. There were no significant differences in cell viability up to 5 days, but long-term exposure to 100 µM of polaprezinc was toxic to hBMSCs (Fig. 1b). However, the addition of the same concentration of solvent elicited no cytotoxicity. (However, the addition of 0.5 mM HCl to the culture medium had no cytotoxicity.) Therefore, 50 µM of polaprezinc was the most ideal concentration for this study using hBMSCs. Next, we confirmed the effects of polaprezinc on hBMSC differentiation to osteogenic lineage at the early and late stages of differentiation. The results showed that polaprezinc slightly increased ALP activity in the early stage of hBMSC osteogenesis (Fig. 1c, d), whereas calcium deposition on the late stage of hBMSC osteogenesis was dose-dependently increased by polaprezinc (Fig. 1e, f). Also, we compared the mRNA levels of osteogenic-related genes between vehicle- and polaprezinc-treated hBMSCs. Polaprezinc significantly increased *RUNX2* mRNA levels and downstream genes, such as *ALPL*, *COL1A1*, *SPP1*, *IBSP*, and *BGLAP* (Fig. 1g). Consistently, the RUNX2 protein level was upregulated under polaprezinc treatment, suggesting that polaprezinc might be a positive regulator of hBMSC osteogenesis (Fig. 1h). Therefore, these results indicate that polaprezinc has the potential to enhance hBMSC differentiation into the osteogenic lineage. However, the sole treatment with L-carnosine, one of the compounds chelated in polaprezinc, had no effect on hBMSC osteogenesis (Supplementary Fig. 1a–d).

### Polaprezinc accelerates osteoclast differentiation in mBMMs.
Our previous study showed that osteoclast differentiation was blocked by zinc-sulfate treatment in mBMMs;[12] thus, we expected that polaprezinc might have similar effects when compared to previous results regarding zinc. To test this, we first determined the optimal polaprezinc dose by examining the cytotoxic effects of polaprezinc in mBMMs. No cytotoxicity was observed in any of the groups tested for this study (Fig. 2a); thus, we used 50 µM of polaprezinc, which is same concentration treated in hBMSCs, for further analysis in experiments related to osteoclast differentiation. To clarify the effect of polaprezinc on osteoclast differentiation, mBMMs were induced by treatment with mRANKL in the presence of polaprezinc dose dependently. Unexpectedly, TRAP staining showed that polaprezinc promoted osteoclast formation and fusion (Fig. 2b). Consistently, TRAP activity and the number of multinucleated osteoclasts were enhanced by polaprezinc dose dependently (Fig. 2c, d), indicating that polaprezinc also acts as a positive regulator of osteoclast differentiation in mBMMs. Furthermore, polaprezinc significantly enhanced mRNA levels of *Nfatc1*, *Ctsk*, and *Dcstamp*, which are critical regulators of osteoclast differentiation[26] in mBMMs (Fig. 2e). Consistently, the protein level of NFATc1 was upregulated by

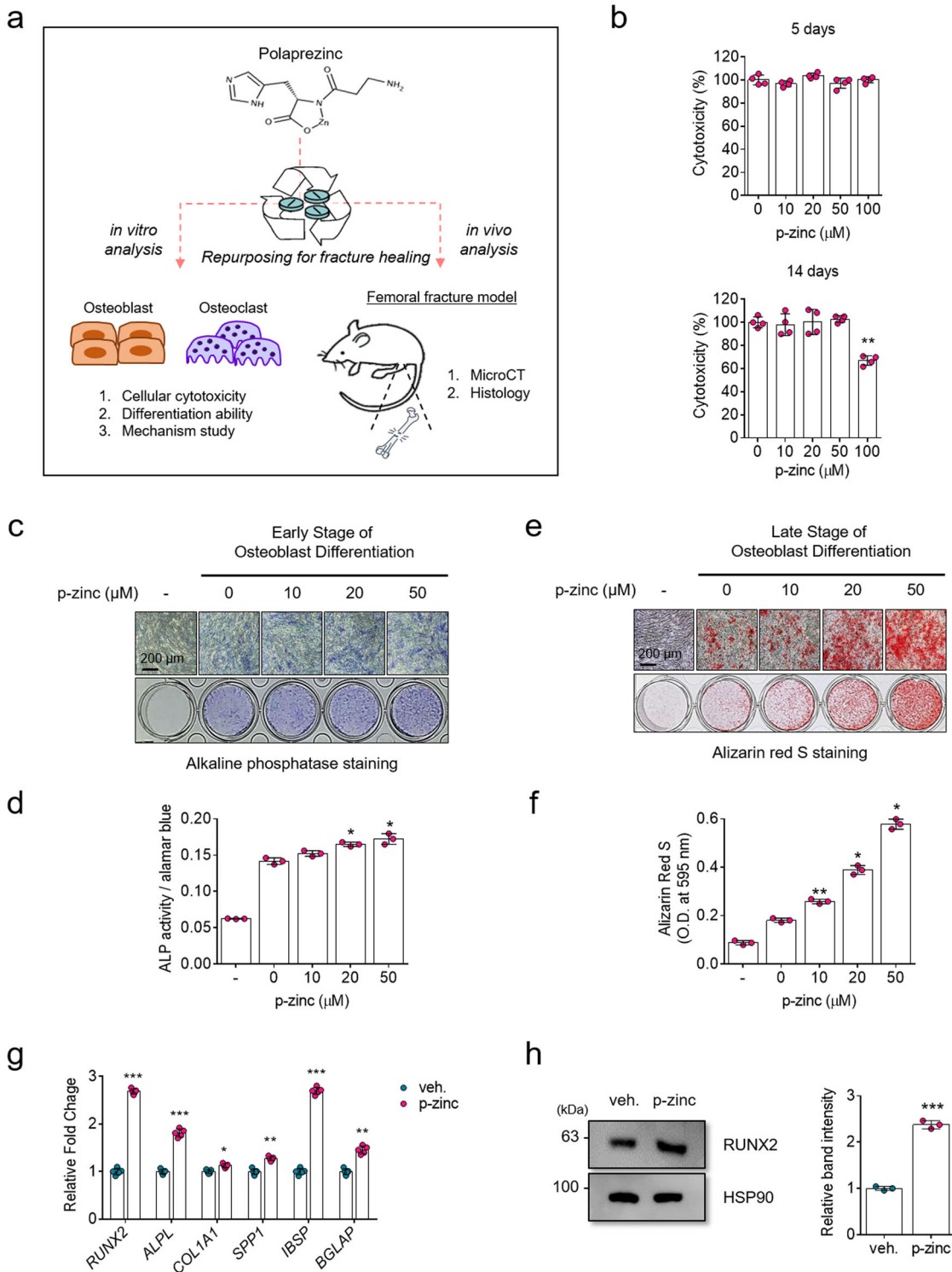

polaprezinc treatment (Fig. 2f). However, standalone treatment with L-carnosine had no effect on osteoclast differentiation (Supplementary Fig. 1e, f), suggesting that the polaprezinc-mediated enhancement of osteoblast and osteoclast differentiation may be due to the enhanced absorption of zinc by L-carnosine[20,21].

**Different effects of polaprezinc and zinc on osteoblast and osteoclast differentiation.** Polaprezinc may have a therapeutic role in bone-related disorders through its dual-positive effects on

osteoclast and osteoblast differentiation. However, we could not explain the difference in molecular mechanisms between polaprezinc and zinc; this clarification is required since zinc inhibits osteoclast activity[27]. To address the difference between polaprezinc and zinc sulfate, we first tested the effects of polaprezinc and zinc sulfate in osteogenesis. Alizarin red S staining showed that more calcium deposits were present in the polaprezinc-treated group (Fig. 3a, b). Also, protein levels of RUNX2 were significantly upregulated in the polaprezinc-treated group, compared to the zinc-sulfate group (Fig. 3c). In mBMMs, TRAP

**Fig. 1 Schematic of the study and effects of polaprezinc on the osteogenic differentiation of hBMSCs. a** Human bone marrow-derived mesenchymal stem cells (hBMSCs) and mouse bone marrow-derived monocytes (mBMMs) were used for in vitro studies. For animal studies, ICR mice aged 8 weeks were used for further in vivo analysis. Polaprezinc was daily administered by oral gavage. **b** The Ez-Cytox assay was used to determine the cellular toxicity of polaprezinc-treated hBMSCs. Each experiment was performed in triplicate ($n = 3$). **$P < 0.01$ compared with vehicle-treated hBMSCs. **c** hBMSCs treated with vehicle or polaprezinc (10, 20, and 50 μM) were incubated in osteogenic medium for 7 days. ALP staining was performed to determine the extent of the initial differentiation at day 7. Scale bar = 200 μm. **d** Graphs were generated from real-time quantitative PCR data using RNA extracted from hBMSCs treated with polaprezinc (50 μM) under osteogenic conditions for 3 days. The results were compared to those of the group treated with vehicle ($n = 3$ experimental replicates). **e** Alizarin red S staining was performed to detect mineral deposition on day 14. Scale bar = 200 μm. **f** The ALP activity assay was performed for the quantitative analysis of ALP staining. The absorbance was measured at 405 and 450 nm and normalized to that of alamar blue staining. **g** For quantitative analysis of alizarin red S staining, absorbance was measured at 595 nm following destaining with 10% cetylpyridinium for 30 min. *$P < 0.05$ and **$P < 0.01$ compared with vehicle-treated hBMSCs. **h** The protein level for RUNX2 was analyzed in hBMSCs using western blotting for cells treated with vehicle or polaprezinc (50 μM) under osteogenic conditions for 5 days. The band intensity was quantified using ImageJ software ($n = 3$, in triplicate) and each RUNX2 protein level was normalized to the HSP90 protein level. ***$P < 0.001$ compared with vehicle-treated hBMSCs.

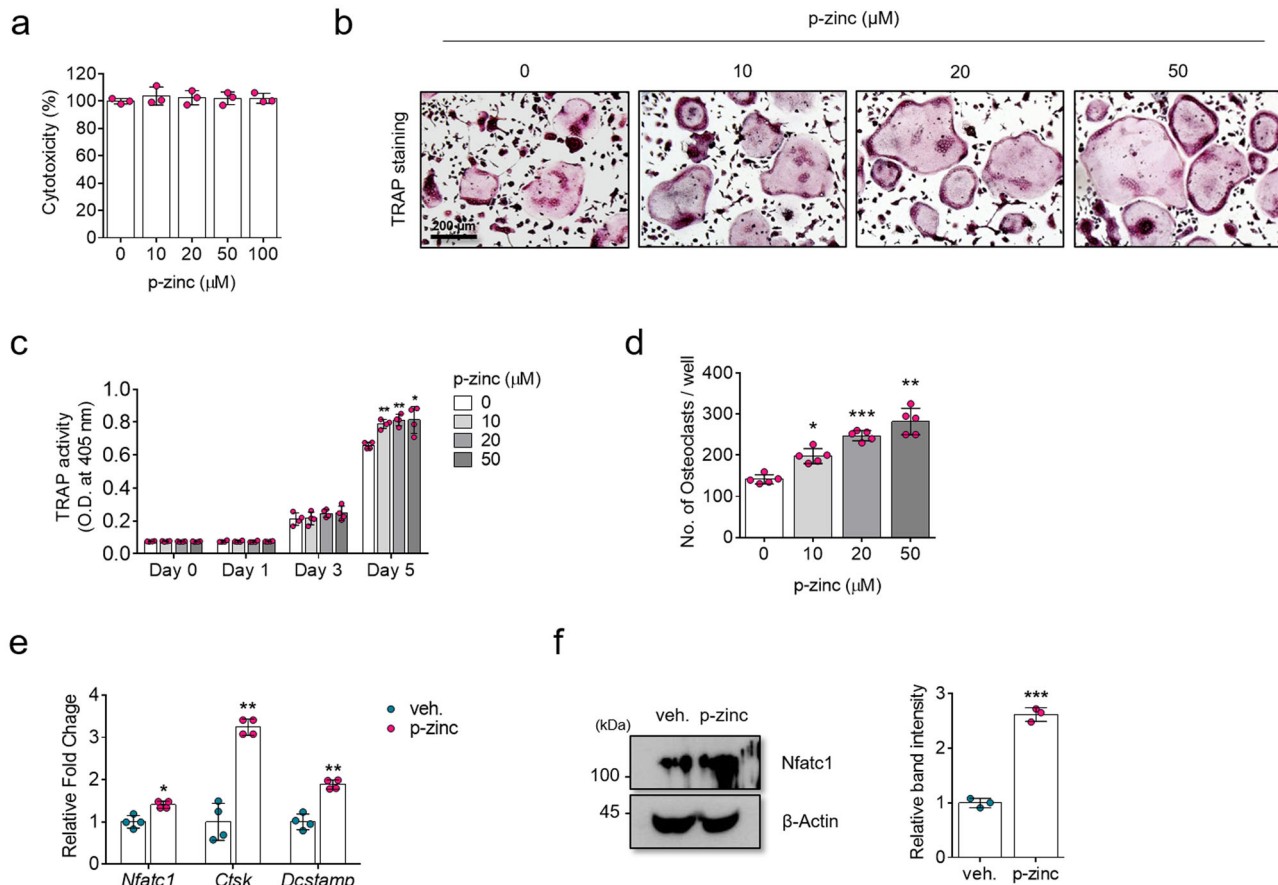

**Fig. 2 Effects of polaprezinc on the osteoclast differentiation of mBMMs. a** The Ez-Cytox assay was used to determine the cellular toxicity of polaprezinc-treated mBMMs at 5 days. Each experiment was performed in triplicate ($n = 3$). There was no significant difference between groups. **b** Representative images of osteoclast differentiation. mBMMs treated with different concentration of polaprezinc were seeded in 12-well culture plates and treated for 5 days with 10 ng/mL M-CSF and 10 ng/mL RANKL. TRAP staining was performed to visualize TRAP-positive mBMMs. Scale bar = 200 μm. **c** TRAP activity was determined as described in the materials and methods section. The absorbance was measured at 405 nm and data are expressed as the mean ± S.D. ($n = 3$) from each independent experiment. *$P < 0.05$ and **$P < 0.01$ compared with vehicle-treated mBMMs. **d** The number of TRAP-positive multinucleated osteoclasts (5 ≥ nuclei) was counted. *$P < 0.05$; **$P < 0.01$; ***$P < 0.001$ compared with vehicle-treated BMMs. **e** Graphs were generated from real-time quantitative PCR data using RNA extracted from mBMMs treated with polaprezinc (50 μM) under osteoclastogenic conditions for 3 days. The results were compared to those of the group treated with vehicle ($n = 3$ experimental replicates). **f** The protein level for NFATc1 was analyzed in mBMMs using western blotting for cells treated with vehicle or polaprezinc (50 μM) under osteoclastogenic conditions for 3 days. The band intensity was quantified using ImageJ software ($n = 3$, in triplicate) and each Nfatc1 protein level was normalized to the β-Actin protein level. ***$P < 0.001$ compared with vehicle-treated mBMMs.

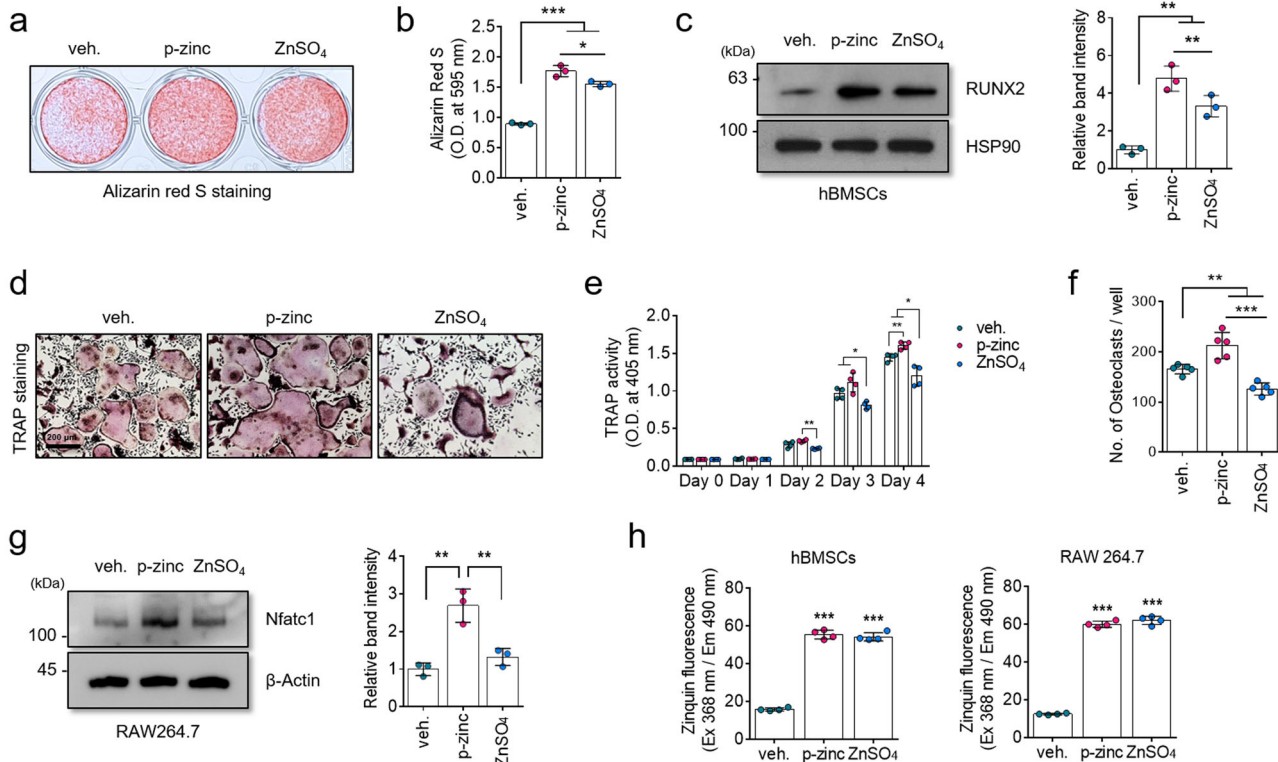

**Fig. 3 Comparative study of polaprezinc and zinc-sulfate differentiation into osteoblast and osteoclast lineages. a** Alizarin red S staining was performed to detect mineral deposition on day 14 in differentiated hBMSCs treated with vehicle, polaprezinc (50 µM), or zinc sulfate (50 µM). **b** For quantitative analysis of alizarin red S staining, absorbance was measured at 595 nm following destaining with 10% cetylpyridinium for 30 min. **c** The protein level for RUNX2 was analyzed in hBMSCs using western blotting for cells treated with vehicle, polaprezinc (50 µM), or zinc sulfate (50 µM). The band intensity was quantified using ImageJ software ($n = 3$, in triplicate) and each RUNX2 protein level was normalized to the HSP90 protein level. $*P < 0.05$; $**P < 0.01$ compared with vehicle-treated hBMSCs. **d** Representative images of osteoclast differentiation. RAW264.7 treated with vehicle, polaprezinc (50 µM), or zinc sulfate (50 µM) were seeded into 12-well culture plates and treated for 5 days with osteoclastogenesis-related reagents. TRAP staining was performed to visualize TRAP-positive RAW264.7. Scale bar = 200 µm. **e** TRAP activity was measured at 405 nm, and the data are expressed as the mean ± S.D. ($n = 3$) from each independent experiment. **f** The number of TRAP-positive multinucleated osteoclasts (5 ≥ nuclei) was counted. **g** The protein level for NFATc1 was analyzed in RAW264.7 using western blotting for cells treated with vehicle, polaprezinc (50 µM) or zinc sulfate (50 µM). The band intensity was quantified using ImageJ software ($n = 3$, in triplicate) and each Nfatc1 protein level was normalized to the β-Actin protein level. $**P < 0.01$ compared with vehicle-treated RAW264.7. **h** Cells treated with vehicle, polaprezinc (50 µM), or zinc sulfate (50 µM) were incubated with 25 µM of Zinquin for 30 min at 37 °C. The fluorescence intensity was measured at excitation 368 nm and emission 490 nm using a fluorometer.

staining showed that polaprezinc enhanced osteoclast activity, which was decreased in the zinc-sulfate group (Fig. 3d). Consistently, TRAP activity and the number of multinucleated osteoclasts showed similar results (Fig. 3e, f). Moreover, protein levels of NFATc1 were upregulated in the polaprezinc-treated group, but downregulated in the zinc-sulfate group (Fig. 3g). Additionally, hBMSCs and mBMMs showed no significant differences in zinc absorption between polaprezinc and zinc sulfate (Fig. 3h). Thus, we concluded that polaprezinc had a dual-positive effect on the differentiation of osteoblasts and osteoclasts, which differed from its role with regular zinc.

**YAP may be a responsible molecular mediator of polaprezinc-induced osteoblast and osteoclast activity.** YAP is a transcriptional coactivator that regulates various cellular mechanisms and itself can be regulated during osteoblast and osteoclast differentiation[28,29]. YAP activates the transcription of NFATc1 by interacting with TEAD4 and activator protein-1 (AP-1) to positively regulate osteoclastogenesis[29]. The role of YAP in osteogenesis is controversial in terms of its regulation of RUNX2 activity, but a recent study clarified the ambiguous role of YAP in regulating osteogenesis[30]. They found that YAP can indirectly activate RUNX2 activity by interacting with AP 2a through a

negative feedback mechanism. Overall, YAP has is a common regulator capable of upregulating the activity of the RUNX2 and NFATc1 proteins; thus, we hypothesized that YAP could be responsible for the polaprezinc-mediated upregulation of both proteins. To test this hypothesis, we first observed YAP protein levels in hBMSCs and RAW264.7 cells treated with polaprezinc or zinc sulfate. Interestingly, total YAP protein levels were upregulated in both polaprezinc- and zinc sulfate-treated hBMSCs and RAW264.7 cells (Fig. 4a). In hBMSCs, both compounds elevated YAP protein levels, which is consistent with the results of hBMSC osteogenesis enhanced by polaprezinc and zinc sulfate. Both compounds also elevated the protein levels of YAP in RAW264.7 cells treated with polaprezinc and zinc sulfate. However, this result was inconsistent with those of mBMM osteoclastogenesis because upregulation of YAP positively contributes to osteoclastogenesis, as mentioned previously. Thus, we hypothesized that zinc sulfate, unlike polaprezinc, has a different mechanism for regulating the transcriptional activity of the YAP protein on osteoclastogenesis. Phosphorylation of the YAP protein leads to its cytoplasmic retention[31], thus shutting off nuclear translocation, leading to loss of transcriptional activity regardless of the increase in the amount of protein. Our western blot results regarding YAP phosphorylation may indicate that polaprezinc

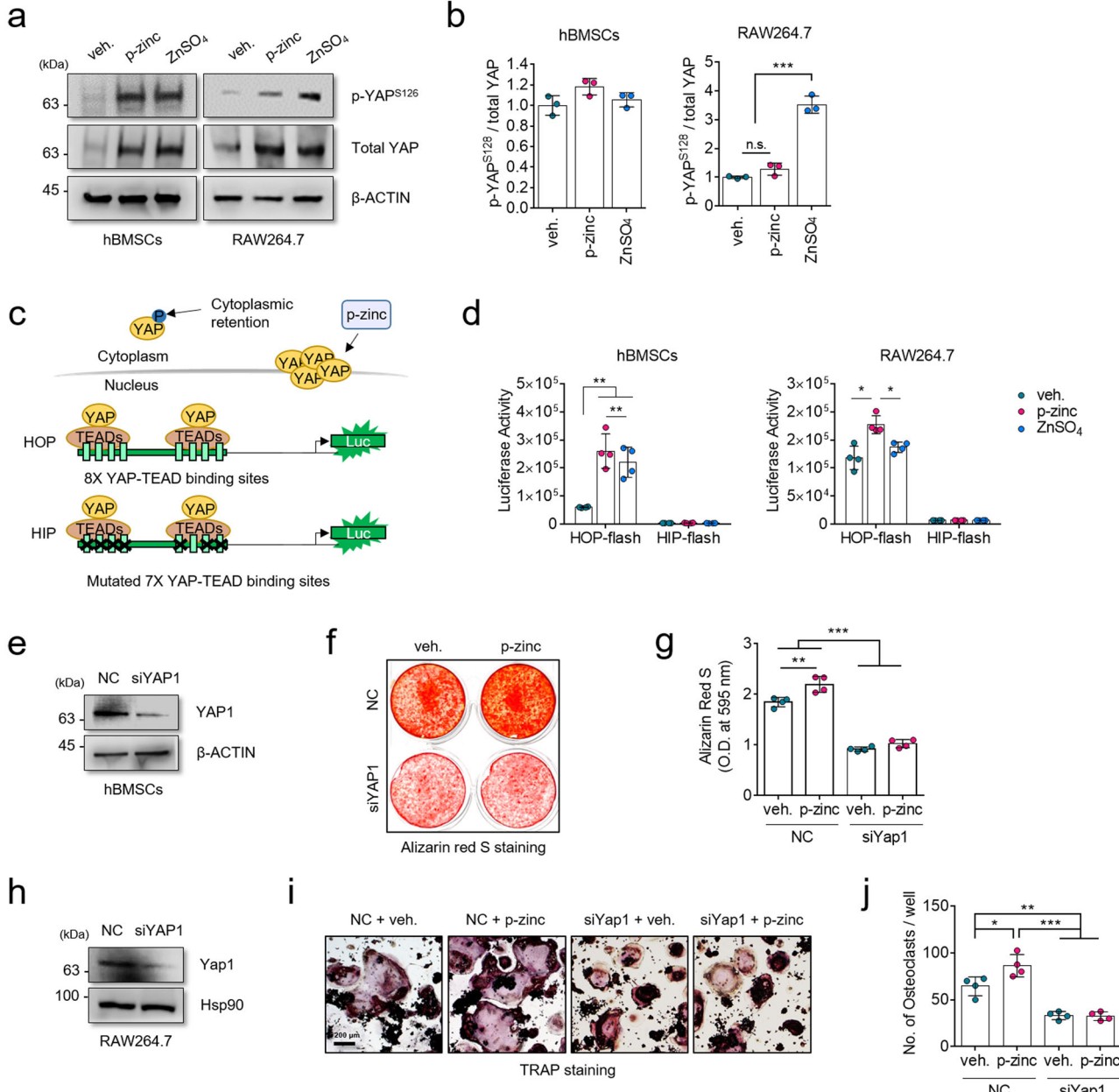

**Fig. 4 Knockdown study of *YAP* in polaprezinc-treated hBMSCs and RAW264.7 cells. a** Protein levels of YAP and phosphorylated YAP$^{S126}$ were analyzed in hBMSCs and RAW264.7 cells using western blotting for cells treated with vehicle, polaprezinc (50 µM), or zinc sulfate (50 µM). **b** The band intensity was quantified using ImageJ software (*n* = 3, in triplicate) and the protein level of phosphorylated YAP$^{S126}$ was normalized to total YAP protein level. ***$P < 0.001$ compared with vehicle-treated cells. **c** Schematic of the HOP/HIP-flash reporter assay. The HOP luciferase vector has eight YAP-TEAD binding sites whereas seven YAP-TEAD binding sites are mutated in the HIP vector. The quantitative increase of YAP protein by polaprezinc is expected to increase the transcriptional activity of YAP protein. **d** YAP-TEAD transcriptional activity was assessed by luciferase reporter constructs in hBMSCs or RAW264.7. *$P < 0.05$; **$P < 0.01$. **e** Western blot analysis of YAP1 protein levels upon transfection with or without *YAP1* siRNA in hBMSCs. **f** Alizarin red S staining was performed to detect mineral deposition on day 14 in negative control or human YAP-targeting siRNA-transfected hBMSCs treated with vehicle or polaprezinc (50 µM). **g** For quantitative analysis of alizarin red S staining, absorbance was measured at 595 nm following destaining with 10% cetylpyridinium for 30 min. **h** Western blot analysis of Yap1 protein level transfected with or without *Yap1* siRNA in RAW264.7. **i** Negative control or mouse YAP-targeting siRNA-transfected RAW264.7 cells treated with vehicle or polaprezinc (50 µM) were seeded into 24-well culture plates and treated for 5 days with RANKL (50 ng/mL). TRAP staining was performed to visualize TRAP-positive cells. Scale bar = 200 µm. **j** The number of TRAP-positive multinucleated osteoclasts (5 ≥ nuclei) was counted.

increased the nuclear localization of YAP protein in RAW264.7 cells, whereas zinc-sulfate upregulated YAP phosphorylation (Fig. 4a, b). To clarify this, we employed the HOP/HIP-flash reporter assay to determine whether polaprezinc affects the transcriptional activity of YAP (Fig. 4c). The transcriptional activity of YAP protein in hBMSC was increased in the

polaprezinc-treated group and the zinc sulfate-treated group, confirming that both compounds are involved in enhancing the transcriptional activity of the YAP protein (Fig. 4d). However, the transcriptional activity of YAP in RAW164.7 cells was increased only in the polaprezinc-treated group (Fig. 4d) despite increases in the amount of YAP protein in the zinc-sulfate-treated group

(Fig. 4a). To address the action mechanism of polaprezinc, which regulates osteoblast and osteoclast differentiation, we performed siRNA-mediated gene knockdown studies targeting *YAP*. The knockdown efficiency of YAP was confirmed by Western blot analysis (Fig. 4e, h). The siRNA-mediated knockdown of *YAP* suppressed the osteogenic potential of hBMSCs, as well as the enhancement of the polaprezinc-mediated osteogenic potential (Fig. 4f, g). Likewise, *YAP* knockdown decreased the osteoclastogenic potential of RAW264.7 cells, and the effect of polaprezinc in the enhanced osteoclast differentiation was abrogated by *YAP* knockdown (Fig. 4i, j). Thus, we concluded that the polaprezinc-mediated enhancement of osteoblast and osteoclast differentiation depends on the presence of YAP, especially in its transcriptional activity.

**Oral administration of polaprezinc accelerates bone remodeling in a mouse fracture model**. To evaluate whether polaprezinc may serve as a supplement for enhancing the treatment of fracture healing, mice with femoral fractures were employed for animal studies in vivo. In a previous study, Yamaguchi and Ozaki reported that oral administration of 25 mg/kg of polaprezinc significantly increased zinc and calcium contents in femoral diaphysis, as well as alkaline phosphatase activity, in weanling rats[32]. We also employed the same dosage of polaprezinc for the current study. From the day after surgery, PBS or polaprezinc was administered daily by oral gavage, and mice were euthanized for further analysis 21 days after surgery (Fig. 5a). µCT imaging revealed that the polaprezinc group had lower remaining callus volume than the PBS group (Fig. 5b). However, callus BMD was significantly increased in the mice orally administered polaprezinc (Fig. 5c), indicating that polaprezinc had a positive effect on bone fracture healing. Safranin O staining was conducted to evaluate the osseous calluses formation at the time of euthanasia. The PBS group showed remnant cartilaginous calluses, whereas polaprezinc group showed new bone formation with less cartilage tissue in the fracture areas (Fig. 5d), implying that cartilaginous calluses were already replaced by new bone tissue in the mice orally administrated polaprezinc. In quantification of callus regions, cartilage areas and fibrotic tissue areas were significantly reduced; however, bone areas were larger in the polaprezinc group than the PBS group (Fig. 5e). These results indicate that polaprezinc promotes cartilage to bone transformation in fractured callus. Next, TRAP staining was performed to evaluate the number of osteoclasts in the fracture areas. The results showed that the number of osteoclasts increased in the fracture areas of the mice administered polaprezinc (Fig. 5f, g). To further investigate the mechanism of accelerated bone healing in the mice administered polaprezinc, we performed immunohistochemistry analysis in fractured calluses. The polaprezinc group showed considerably more osteocalcin protein in the fracture areas of the bone lining cells. Also, YAP was detected in new bone area in the callus and upregulated in the polaprezinc group (Fig. 5h, i). These results indicate that oral administration of polaprezinc can induce rapid and successful fracture healing in mice with femoral fractures through active bone homeostasis by multiple osteoblasts and osteoclasts.

## Discussion
Despite advanced technology and scientific knowledge on treating many types of human diseases, the development of new drugs is extremely expensive and time consuming for such technologies to be applied to patients[33]. These processes place a huge burden on both pharmaceutical companies and patients. Therefore, minimizing time-consuming and costly processes can be a prerequisite for the pharmaceutical industry and inventors. Drug repositioning is a useful strategy for discovering new indications for existing drugs that are inexpensive and risk-free. The development of new drugs involves

very complex processes, whereas drug repositioning has only four steps, compound identification, compound acquisition, development, and FDA post-market safety monitoring[34]. Zinc is a useful biomolecule for applications in fracture healing and bone tissue engineering[35]. Nevertheless, there is a need for further studies regarding whether zinc can be used as an effective supplement for bone defect healing in patients with fractures[4,19]. In the fracture healing process, osteoclasts play an important role in the formation of new bones. In this process, the apoptosis of hypertrophic chondrocytes and resorption of the mineralized cartilage matrix by osteoclasts occurs, and is then replaced by new bone at the site of absorption[36–38]. Therefore, the active roles of both osteoblasts and osteoclasts are required for a successful fracture healing process. However, it has long been considered that zinc inhibits the activity of osteoclasts[27,39,40]. If osteoclast activity continues to be suppressed during fracture healing, it can interfere with the normal process of new bone formation. In this regard, polaprezinc is an ideal supplement for patients with fractures.

In the current study, we showed that polaprezinc positively regulates osteoblast and osteoclast differentiation by upregulating mRNA and protein levels of RUNX2 and NFATc1, which are master regulators of osteoblast and osteoclast differentiation[41,42]. This suggests the possibility of using zinc as a novel dual-positive therapeutic agent against bone fracture, osteoporosis, and other bone diseases. The risk of bone-related diseases is increased by an imbalance in bone resorption and bone formation by osteoclasts and osteoblasts[43]. In the process of bone healing or remodeling, osteoclasts primarily resorb damaged bone, allowing osteoblasts to further restore the shape and structure of the bone;[44] thus, osteoclasts play a pivotal role during fracture healing. A strategy that disrupts pharmacological osteoclast activity has been used as a therapy for post-menopausal bone loss, but fracture studies using such pharmacological mechanisms in genetic knockout mice have encountered enlarged malformed calluses and persistent fracture gaps[45], demonstrating the essential role of osteoclasts in soft-callus formation and healing processes. Therefore, maintaining the activity of osteoclasts in vivo during fracture healing would be beneficial.

Mechanistically, we proved that polaprezinc treatment induced nuclear localization of YAP protein. Nuclear YAP can initiate NFATc1 transcription by binding AP-1 and TEAD to regulate osteoclastogenesis and related gene expression[29]. In addition, the YAP/RUNX2 axis contributes to successful osteogenesis[46,47]. Recent studies conducted by the Boerckel JD group proved the possibilities of YAP/TAZ as potential therapeutic targets of bone homeostasis and fracture healing. They showed that YAP and TAZ coordinately enhanced osteoblast activity and osteoclast-mediated bone remodeling to promote bone development[48]. In addition, YAP appears to be involved in the expansion and differentiation of periosteal osteoblastic precursor cells in order to promote bone fracture healing[49]. Interestingly, YAP has been shown to be associated with the osteocyte-mediated bone remodeling processes by regulating the mechanical properties of bone matrix organization[50]. Ultimately, evidence from recent studies suggests that YAP-related signaling pathways are deeply connected to skeletal development, as well osteoblast/osteoclast interactions and osteocyte-mediated bone remodeling[51]. Therefore, developing drugs that can specifically activate YAP-related signaling pathways may be a strategy with which to treat bone development- or bone fracture-related injuries. In this study, we showed that zinc or polaprezinc treatment upregulate protein levels of YAP in hBMSCs and RAW264.7 cells. Oral administration of polaprezinc increased YAP-positive cell numbers in the fracture calluses of experimental animals. Considering the results from recent studies, it is clear that zinc is a good candidate for bone homeostasis. Furthermore, our study demonstrated a relationship between zinc and YAP in osteoblast and osteoclast

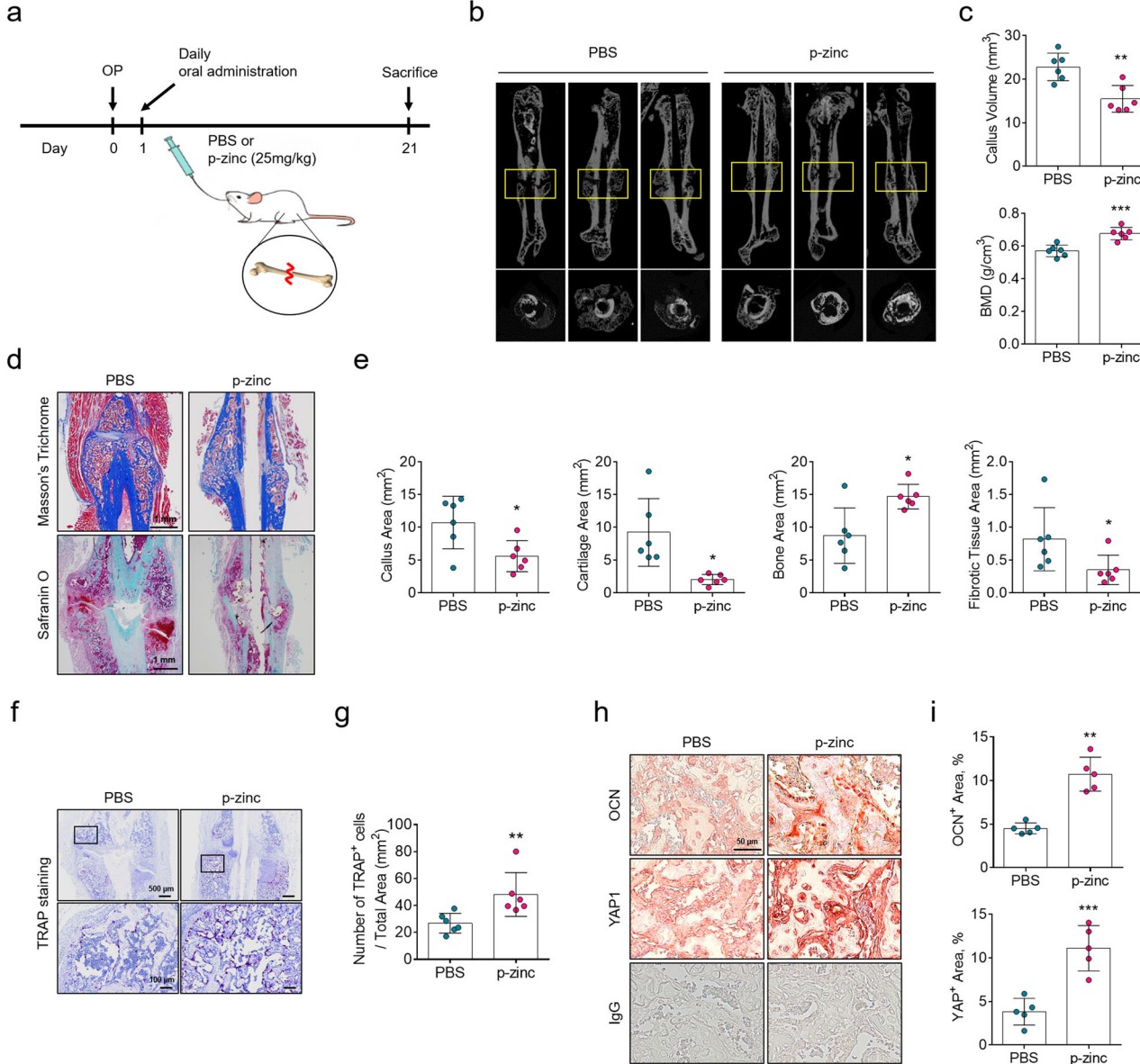

**Fig. 5 Micro-CT analysis in a mouse femoral fracture model and the effects of polaprezinc. a** Scheme of the animal study. PBS or polaprezinc (25 mg/kg) was orally administered daily after fracture. Male ICR mice at 8 weeks of age were euthanized at 21 days post-procedure ($n = 6$). Femoral samples from each group were collected and subjected to µCT analysis to examine callus formation. **b** Representative µCT images of fracture calluses at 3 weeks post-closed femur fracture. **c** Callus volume and bone mineral density (BMD) were quantified using µCT analysis. **d** Representative images of the fracture callus stained with safranin O. Scale bar = 1 mm. **e** Quantification of callus area, cartilage area, bone area and fibrotic tissue area. **f** Representative images of the fracture callus stained with TRAP. Scale bar = 500 or 100 µm. **g** Quantification of TRAP-positive cells from total cell population per total area (mm²) in histological sections. **h** Representative images of the fracture callus stained with anti-osteocalcin or anti-YAP1 antibody. Scale bar = 50 µm. **i** Quantification of OCN or YAP-positive area by ImageJ software.

differentiation. In particular, polaprezinc treatment increased the transcriptional activity of YAP protein in hBMSCs and RAW264.7 cells more than that of the zinc-sulfate treatment, although we have not provided detailed mechanisms for how polaprezinc or zinc may be involved in YAP protein increases. Nevertheless, the current study is meaningful in that it is the first study to show that polaprezinc is involved in the differentiation of osteoblasts and osteoclasts by enhancing the transcriptional activity of YAP.

Collectively, our results regarding the effects of polaprezinc on hBMSC osteogenesis and mBMM osteoclastogenesis provide new insights into successful fracture healing by maintaining healthy osteoblasts and osteoclasts throughout. Therefore, we believe that polaprezinc may not only be a supplement for successful fracture

healing but also a good candidate for clinical application through drug repositioning.

## Methods

**Cell culture and reagents**. This study was approved by the Institutional Review Board (IRB) of Yonsei University College of Medicine (IRB No: 4-2017-0232) and informed consent was obtained from all participants for human bone marrow samples used in this study. Bone marrow aspirates were obtained from the posterior iliac crests of seven adult donors (55–65 years old) and isolated as previously described[52]. hBMSCs were maintained in low-glucose Dulbecco's modified Eagle's medium (DMEM-LG; Gibco) supplemented with 10% FBS (Gibco) and 1% antibiotic/antimycotic solution (Gibco) and were used within five passages. RAW264.7 cells (Korean Cell Line Bank, Seoul, South Korea) were maintained in high-glucose DMEM (Gibco) supplemented with 10% FBS (Gibco) and 1% antibiotic/antimycotic solution (Gibco). mBMMs were cultured in α-minimum essential medium

**Table 1 Primers sequences used for qRT-PCR.**

| Gene | Primer sequence (5′ → 3′) |
|---|---|
| RUNX2 | F: TACAAACCATACCCAGTCCCTGTTT |
|  | R: AGTGCTCTAACCACAGTCCATGCA |
| COL1A1 | F: GCCCTGCTGGAGAGGAAGGA |
|  | R: GCCAGGGAAACCACGGCTAC |
| SPP1 | F: CCGTTGCCCAGGACCTGAA |
|  | R: TGTGGCTGTGGGTTTCAGCA |
| IBSP | F: ATACCATCTCACACCAGTTAGAATG |
|  | R: AACAGCGTAAAAGTGTTCCTATTTC |
| BGLAP | F: AGAGCCCCAGTCCCCTACCC |
|  | R: AGGCCTCCTGAAAGCCGATG |
| GAPDH | F: CTGCTGATGCCCCCATGTTC |
|  | R: ACCTTGGCCAGGGGTGCTAA |
| Nfatc1 | F: GGTAACTCTGTCTTTCTAACCTTAAGCTC |
|  | R: GTGATGACCCCAGCATGCACCAGTCACAG |
| Dcstamp | F: TCCTCCATGAACAAACAGTTCCAA |
|  | R: AGACGTGGTTTAGGAATGCAGCTC |
| Ctsk | F: AGGCAGCTAAATGCAGAGGGTACA |
|  | R: ATGCCGCAGGCGTTGTTCTTATTC |
| Gapdh | F: GTGTTCCTACCCCCAATGTGT |
|  | R: ATTGTCATACCAGGAAATGAGCTT |

(Gibco) containing 10% FBS and 1% antibiotic-antimycotic solution. Polaprezinc (CAS 107667-60-7, Santa Cruz Biotechnology, Santa Cruz, CA, USA) was dissolved in 1 M hydrogen chloride and used at a final concentration of 50 μM. $ZnSO_4$ was purchased from Sigma-Aldrich.

**Cell viability measurement**. Cell viability was assessed to measure the cytotoxicity of polaprezinc. Briefly, the cells were maintained in growth media and the media were changed every 2 days with different concentrations of polaprezinc or $ZnSO_4$ for 14 days. To measure the levels of cell viability, Ez-Cytox (DoGen, Seoul, Korea) reagent was added to each well and incubated for 2 h at 37 °C. Cell viability was measured at 450 nm using a microplate reader (VersaMax™ Microplate Leader, CA, USA).

**In vitro osteogenic differentiation**. hBMSCs were seeded at $8 × 10^4$ cells/well in 12-well plates. To induce in vitro osteogenesis, hBMSCs were cultured in osteogenic medium [DMEM-LG containing 10% FBS, 1% antibiotic-antimycotic solution, 10 mM β-glycerophosphate (Sigma-Aldrich), and 50 μg/mL ascorbic acid (Gibco)] with or without polaprezinc or $ZnSO_4$ for 5–12 days. Polaprezinc- or $ZnSO_4$-containing osteogenic medium was replaced every other day.

**Alkaline phosphatase and alizarin red S staining**. To confirm the differentiation activity of osteogenesis, we performed alkaline phosphatase (ALP) and alizarin red S staining. For ALP staining, the cells were fixed with citrate buffer:acetone fixative (2:3). hBMSCs were stained using alkaline staining solution, which was mixed with fast blue RR salt (Sigma-Aldrich) and naphthol AS-MX phosphate alkaline solution (Sigma-Aldrich) for 30 min. To measure ALP activity, hBMSCs were incubated with substrate solution containing 0.5 M $Na_2CO_3$, 0.5 M $NaHCO_3$, 1 M $MgCl_2$, and phosphatase substrate (Sigma-Aldrich) for 30 min at room temperature. The absorbance of the substrate solution was measured at 405 and 450 nm. ALP activity was normalized to Alamar blue (Invitrogen, Carlsbad, CA, USA). For alizarin red S staining, differentiated hBMSCs were fixed in ice-cold 70% ethanol and stained with 2% alizarin red S solution. For quantification of alizarin red S staining, the cells were stained with 10% cetylpyridinium chloride and the absorbance was measured at 595 nm using a microplate reader.

**In vitro osteoclastogenic differentiation**. Preparation of mBMMs was previously described[12]. Briefly, mouse bone marrow-derived cells were cultured in growth media. After 48 h, non-adherent cells were collected and cultured with 10 ng/mL mM-CSF (R&D Systems, Minneapolis, MN, USA). mBMMs were detached using Detachin™ (Genlantis, San Diego, CA, USA) and $5 × 10^4$ cells were seeded on 24-well plates for osteoclastogenesis. mBMMs were cultured in growth media containing 10 ng/mL mM-CSF and 10 ng/mL mRANKL (R&D Systems) with or without polaprezinc. In addition, for inducing osteoclastic differentiation, RAW264.7 cells were plated at a concentration of 10,000 cells/well in a 24-well plate and treated with 50 ng/mL mRANKL in growth media with or without polaprezinc or $ZnSO_4$.

**TRAP activity and TRAP staining**. To evaluate TRAP activity, 50 μL of osteoclast culture supernatant was incubated with a substrate mix containing acetate solution (Sigma-Aldrich), 1 M sodium tartrate, and phosphatase substrate (Sigma-Aldrich)

for 1 h at 37 °C. The reaction was stopped by adding 3 N HCl and measured at an absorbance wavelength of 405 nm. TRAP staining was performed using an Acid Phosphatase, Leukocyte kit (Sigma-Aldrich) following the manufacturer's instructions. Cells were fixed with fixative solution and stained with staining solution for 1 h at 37 °C. TRAP-positive multinucleated cells containing more than five nuclei were considered as osteoclasts.

**Quantitative real-time polymerase chain reaction (qRT-PCR)**. Total RNA was extracted using the RNeasy Mini Kit (Qiagen, Hilden, Germany) according to the manufacturer's instructions. Complementary DNA was synthesized using the Omniscipt Reverse-Transcription Kit (Qiagen) following the manufacturer's protocol. qRT-PCR was performed using 2 × qPCRBIO SyGreen Mix Hi-Rox (PCR Biosystems, London, UK). The specific primer sets (Bioneer, Daejeon, Korea) are shown in Table 1. *ALPL* (P324388) primer sets were purchased from Bioneer.

**Western blot analysis**. For immunoblotting, cells were lysed with ProPrep™ (iNtRON Biotechnology, Seongnam, Gyeonggi-do, Korea). Whole cell lysates were centrifuged at 13,000 rpm for 20 min at 4 °C and quantified using a SMART™ BCA Protein Assay Kit (iNtRON). Protein samples were loaded onto SDS-PAGE gels and transferred to a polyvinylidene difluoride membrane (Amersham Pharmacia, Piscataway, NJ, USA). Western blot analysis was performed using primary antibodies against mouse anti-RUNX2 (1:1000; Millipore, Burlington, MA, USA), rabbit anti-HSP90 (1:5000; Santa Cruz Biotechnology), rabbit anti-NFATc1 (1:1000; Santa Cruz Biotechnology), anti-yes-associated protein-1 (1:1000; YAP; Cell Signaling), anti-phosphoYAP$^{S126}$ (1:1000; Cell Signaling), and mouse anti-β-ACTIN (1:5000; Santa Cruz Biotechnology). ECL solution (Bio-Rad, Hercules, CA, USA) was used to visualize the signals. Relative band intensities were measured using ImageJ software (National Institutes of Health).

**Intracellular zinc measurement**. Zinquin (Sigma) was used to detect intracellular zinc levels after treatment with polaprezinc and $ZnSO_4$. Briefly, polaprezinc and $ZnSO_4$ were applied to hBMSCs or RAW264.7 cells. After 24 h, 25 μM of Zinquin was added and incubated for 30 min at 37 °C, and cells were washed three times with PBS to remove extracellular Zinquin. Fluorescence was measured at an excitation wavelength of 368 nm and emission of 490 nm using a Fluorometer (Varioskan Flash 3001, Thermo Fisher Scientific, Waltham, MA, USA).

**Luciferase assay**. Cells were seeded in triplicate in six-well culture plates at $1 × 10^5$ cells and cultured for 24 h, and the luciferase reporter assay was performed as previously described[53]. The cells were transfected with 500 ng of HOP-Flash (83467, Addgene) or HIP-Flash luciferase reporter plasmid (83466, Addgene) along with 10 ng of pRL-TK Renilla plasmid (Promega) using the Neon Transfection System (Invitrogen) according to the manufacturer's instructions. The day after transfection, cells were treated with 50 μM polaprezinc or $ZnSO_4$, respectively. Luciferase and Renilla signals were measured 48 h after transfection using a Dual Luciferase Reporter Assay Kit (Promega, Madison, WI, USA) according to the manufacturer's instructions.

**siRNA-mediated knockdown**. Synthetic small interfering RNAs (siRNAs) for human and mouse *YAP* and non-targeting siRNAs were purchased from Bioneer. Each siRNAs were transfected using the Neon Transfection System (Invitrogen) following the manufacturer's protocol. Briefly, 100 pmol of siRNAs per $5 × 10^5$ cells were transfected under the manufacturer's parameters (hBMSCs: 990 V, 40 ms, 1 pulse, 100 μl tip; RAW264.7: 1680 V, 20 ms, 1 pulse, 100 μl tip). After microporation, cells were transferred to 6-well plates containing complete media without antibiotics. After 48 h, YAP1 expression levels were determined by Western blot analysis.

**Mouse femoral fracture model**. All animal experiments were performed in accordance with the Institutional Animal Care and Use Committee (IACUC) and approved by the Yonsei University College of Medicine (Permit No: 2018-0146). The mice were maintained under a 12 h light/dark cycle, 22 ± 2 °C, and 50 ± 5% humidity. They had ad libitum access to food and water. A standardized mid-diaphyseal fracture was induced in male ICR mice at 8 weeks of age. Mice were anesthetized with Zoletil (30 mg/kg, Virbac, Carros, France) and Rompun (10 mg/kg, Bayer, Ontario, Canada) by intraperitoneal injection. The left leg was shaved, and an anterior knee incision was made. The vastus lateralis was elevated to expose the femur, and transverse osteotomy was performed. The fractured femur was fixed using a 22-gauge needle. The intramuscular wound and skin incision were closed using nylon suture. Subcutaneous injection of Metacam (1 mg/kg, Boehringer Ingelheim, Ingelheim am Rhein, Germany) was administered immediately following surgery for pain management. Also, mice received pain management until post-surgical day 3. The mice with fractured limbs were randomly divided into the two following groups: vehicle-treated and polaprezinc-treated. Vehicle and polaprezinc (25 mg/kg) were ~~orally~~ administered daily after fracture using a feeding catheter (18G × 37 mm, C1LifeTECH, Cheongju, Chungcheongbuk-do, Korea). Mice were euthanized 21 days post-procedure ($n = 6$ per group).

**Micro-computed tomography (µCT)**. For µCT analysis, femoral samples were fixed in 70% ethanol for 24 h at room temperature. The fixed samples were analyzed using high-resolution µCT (Skyscan-1173, Skyscan, Kontich, Belgium). µCT image reconstruction and analysis were performed using the reconstruction software NRecon (v1.6.9.8, Skyscan) and CT-analyzer software CTAn (v1.13.2.1, Skyscan), respectively. Epiphyseal trabecular bone measuring parameters were analyzed using the 3D model visualization software CT-Vol (v2.0, Skyscan). The acquisition setting conditions were followed by an X-ray source voltage of 90 kVp and current of 88 µA. Beam hardening reduction depended on a 1.0-mm-thick aluminum filter. The pixel size was 7 µm, exposure time was 500 ms, the rotation step was 0.3°, with full rotation occurring over 360°. The region of analysis was selected as 5 mm from the proximal to the distal of the fracture midline. which is between the intact cortical of the fracture according to Collier et al.[54] The original cortical bone was excluded from callus analysis using CT-Analyzer software. For determination of callus volume, contoured segmentation was applied to the volume of interest, and the polar moment of inertia (pMOI) was calculated. The calluses, both including and excluding intact cortical bone, were contoured and stacked by applying a threshold of 255 mg hydroxyapatite/cm³. Following the µCT measurements, the samples were prepared for histology.

**Immunohistochemical analysis**. The femoral samples from each group were fixed in 10% formalin solution for 5–7 days at room temperature. Samples were decalcified in 0.5 M EDTA (pH 7.4) solution for 2 weeks at room temperature. Decalcified femurs were embedded in paraffin blocks. For paraffin blocks, samples were dehydrated by passage through an ethanol series, cleared twice in xylene, and embedded in paraffin, after which 5 µm sections were cut using a rotary microtome. Decalcified femoral sections were stained with hematoxylin and eosin, Masson's trichrome, safranin O and fast green, and TRAP. For histomorphometric quantification, cartilage areas, bone areas, and fibrotic tissue areas of the fracture site were determined by ImageJ software. At least three non-consecutive sections were used for analysis, and the mean value represented one sample. For immunostaining, the sections were deparaffinized, rehydrated, and antigen retrieval was performed using citrate buffer (pH 6.0). Briefly, sections were blocked with 5% FBS in PBST for 1 h at room temperature and then incubated with anti-osteocalcin (TaKaRa Bio Inc., Shiga, Japan) or anti-YAP1 (Abcam) antibody overnight at 4 °C. After washing in PBST, the samples were incubated with VisUCyte™ HRP polymer antibody (R&D Systems) for 1 h at room temperature. Immunoreactive samples were visualized using an AEC substrate kit (Abcam) as described previously[55]. Isotype IgG antibody was used as a negative control. IHC staining was quantified using ImageJ software[56,57].

**Statistics and reproducibility**. Analyses were performed using one-way ANOVA or Student's $t$ test using GraphPad Prism 6 software. All data are presented as mean ± standard deviation for at least three individual experiments. The animal experiments performed in this study were initially performed using a total of ten mice in each group, but two mice in each group were excluded due to unexpected complications. Thereafter, in histological analysis using eight animals in each group, the values recorded respectively the highest and lowest values were additionally excluded, and finally, statistical analysis was performed using data from six experimental animals in each group (Supplementary Table 1). The histology data were analyzed using the same experimental animals, and the animals excluded from all histological analyzes were identical in all histological graphs.

*Reproducibility*. All experiments in this study include at least three biological replicates, and the number of replicates is mentioned in the text or figure legend.

**Reporting summary**. Further information on research design is available in the Nature Research Reporting Summary linked to this article.

## Data availability
The datasets used and/or analyzed during the current study are included in this published article or available from the corresponding authors on reasonable request. All uncropped blots are included as Supplementary Fig. 2. Source data of the main and Supplementary Figures are available in Supplementary Data 2 (an excel file). Plasmids used in this study were purchased from Addgene [HOP-Flash (83467) or HIP-Flash luciferase reporter plasmid (83466)].

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

## Acknowledgements

This research was supported by Mid-Career Research Program through the National Research Foundation of Korea (NRF) funded by the Ministry of Education (NRF-2019R1A2C1087777).

## Author contributions

E.A.K., Y.J.P., D.S.Y., and K.H.P. conceived and designed the experiments. E.A.K., Y.J.P., D.S.Y., K-M.L., J.K., S.J., and J.W.L. performed the experiments. E.A.K. and D.S.Y. processed and analyzed the data. D.S.Y. wrote initial version of the paper, and E.A.K. and D.S.Y. revised the paper. D.S.Y., K.H.P., and J.W.L. edited the paper. J.W.L. and K.H.P. provided financial support and approved the final version of the paper.

## Competing interests

The authors declare no competing interests.
