## [Peer Review File · Communications Biology]

Reviewers' comments:

Reviewer #1 (Remarks to the Author):

The authors aimed to investigate the effects of polaprezinc (a source for zinc) on fracture healing. The research question is interesting, so are the results. However, the first part of the study (in vitro experiments) are mainly a recapitulation of the author's previous study using ZnSO₄ instead of polaprezinc (which should not make a big difference in vitro, only in vivo). The only new finding is regarding the YAP signaling pathway. In the second part of the study (in vivo experiments), the authors did not evaluate this pathway further. I would strongly recommend to do so to add more novel data to the manuscript and strengthen the research findings.

Major comments

- I am missing a histological quantification of bone, cartilage and fibrous tissue in the fracture callus sections. This is gold standard for fracture healing studies to evaluate endochondral ossification. It would be reasonable to hypothesize, that zinc might also influence cartilaginous cells.
- Osteocalcin staining is not specific for osteocytes in the fracture callus. The authors should rephrase this to osteoblasts.
- The authors propose an involvement of YAP signaling in the effects of zinc on fracture healing. However, they do not show any data regarding this. They should at least stain YAP activation in the fracture callus.

Minor comments:

- line 57: the reference for "zinc deficiency interferes with fracture healing" is missing
- please provide more details about how the fracture was created. Open or closed?
- please describe pain management in more detail. According to international animal ethics, mice should also receive pain management BEFORE a surgery
- line 358: what do the authors mean with 5 mM? Should this be millimeter? Did the authors exclude the former intact cortex from their analysis and only measure the fracture callus?
- abstract: "There are many kinds of medications and supplements available to treat or support patients with fractures" - I don't agree with this sentence, Vit D and calcium are the only broadly available medications/supplementations during fracture healing so far
- abstract: "Therefore, we suggest that drug repositioning of polaprezinc would be helpful for patients with fractures, especially in patients with non-union fractures." The authors did not evaluate the effects of zinc in a non-union model, therefore they should skip the last part of the sentence.

Reviewer #2 (Remarks to the Author):

Ko et al evaluate the role of polaprezinc in bone cell differentiation and in fracture healing. In vitro, pZinc promoted both osteoblast and osteoclast differentiation. This differed from the effects of zinc sulfate, particularly in the setting of osteoclastogenesis. pZinc altered YAP protein expression in vitro which may be the mechanism for its effects. In vivo, animals treated with pZinc showed greater bone formation in their fracture calluses after 21 days.

This work is novel and addresses an area of need, but has some significant flaws and missing information. Even with revisions it only provides very preliminary data on fracture efficacy and more will be needed to seriously pursue repositioning this drug for use in fracture healing.

Major comments

1. The introduction and discussion don't cover all the relevant literature, and the citations appear to be quite selective throughout. Specifically, the discussion of role of zinc in bone homeostasis is very limited. A basic pubmed search turns up multiple studies that assess various types of zinc treatment during bone healing and these are not covered at all. The discussion of the role of YAP

in bone homeostasis and fracture healing is also very limited. Joel Boerkel's lab has published a number of studies addressing this including a recent one looking at fracture healing which may be appropriate to cover. I suggest reworking the introduction and discussion to address this, and to consider addressing the following questions: Does zinc have a specific role in bone formation, such as getting incorporated into the matrix, or being a cofactor for collagen or other ECM synthesis? How common is zinc deficiency, and could this be exacerbated following injury?

2. Not all the conditions of the editorial checklist are met. There is no data availability statement, or ethics statements for animals or humans. The figures should be redone to show individual data points for most of the graphs.

3. The polaprezinc contains covalently bound zinc. The authors imply it is a source of ionic zinc that they expected to be equivalent to zinc sulfate, but this must require some sort of chemical reaction to release the zinc atom. Where does this reaction occur? Will it happen in vitro in the same way as following oral delivery?

4. The osteoclast data should really include treatment with more than one concentration of pZinc (Figure 2), and ideally should have osteoclast numbers per well reported. The data in Figures 3E and 4F in particular have no quantitative measure of osteoclast number which is unacceptable.

5. Figure 3C and 4E have no quantitative measure of mineralization, mineralized area or dye release should be reported.

6. The YAP activity data and its interpretation in Figure 4 is problematic and requires revision. Firstly, the applicability of using Hela cells with exogenous YAP added is limited. Second, the conclusion that p-zinc affects YAP expression cannot be concluded from the reporter data shown. The YAP is expressed from a pcDNA construct so presumably lacks most of its natural promoter, and therefore regulation measured in this system should be at the protein level. It would be more applicable to perform the reporter assay in at least one of the cell types evaluated previously using the naturally occurring YAP expression to confirm if the overall increase in YAP increases its signaling. This figure legend also does not mention transfection with the YAP plasmid.

7. The siRNA studies in Figure 4 lack an siRNA control – it is possible that YAP knockdown affects differentiation in the absence of zinc which is important for understanding the relevance of this knockdown.

8. In vivo study: The fracture parameters reported should include 'total volume' and the overall callus size may change. The histology looking at cartilage vs bone area should also be quantified and presented. Finally the method for counting TRAP+ or OCN+ cells is not mentioned, and the OCN+ cells are called 'osteocytes' in the figure (H) but not in the legend. Depending on the method of analysis, it should be possible to distinguish osteoblasts from osteocytes with relative confidence, but they should only be defined as osteocytes if they are embedded in the bone.

Minor comments:

9. Abstract: the second sentence is not accurate, there are not really approved pharmacotherapies for fracture healing.

10. Introduction – a lot of the first paragraph is unnecessary. The third paragraph also contains excessive explanation of why fractures are a problem/costs involved. The introduction can be reworked as suggested above without changing the length dramatically.

11. Line 51-53: Can the authors be more specific about the effects of zinc on osteoblast and osteoclast differentiation?

12. Section 3.3: there is a lot of unnecessary discussion for a results section in the first few sentences, including unfounded claims about the mechanism of activity in vitro (line 129-130). The reference used to justify the importance of osteoclast activity in fracture healing cites a knockout model where there are defects in both osteoblast and osteoclast differentiation. There are plenty of more appropriate examples of more osteoclast-selective regimes, including antiresorptive treatments. A factor that promotes osteoclastogenesis is potentially a controversial approach, but notably BMP2 promotes osteoclastogenesis in some settings, so it is not without precedent.

13. Results: fracture studies – please state the dosage of polaprezinc and justify the choice of this dose.

14. Fracture study lines 207 and 209 – the data shown does not indicate changes in bone remodelling. Remodelling is a dynamic process and either a time course or histomorphometric measures are required to make this conclusion.

15. Methods: hBMSC protocol should be covered briefly, at least to include ethical permissions, passage number, and how many donors were used.

16. Methods: 1M HCl is very acidic for a vehicle, please clarify the final concentration following

dilution to reassure the readers that it will not affect the final pH.

17. Methods: '10 ng/mL mM of CSF' in line 287 and 290 looks like a typo

18. Methods line 301 – it is unclear what this statement about TRAP+ cells means, especially as osteoclast counts are not actually presented in the manuscript

19. Zinquin protocol lines 323-327: was there a washing step in this protocol?

20. Luciferase assay: a plasmid that expresses Renilla luciferase is not mentioned but is used as a control.

21. Fracture model: please describe more thoroughly how the fracture is generated, and specify the method of oral delivery.

22. MicroCT methods: Please double check the description from line 358-360. How was the centre of the fracture identified, and how was the original cortical bone excluded from analysis?

23. Methods: please double check the use of mM vs mm.

24. In vivo study – it is not clear if the same bones or different ones are used for microCT and histology. N=10 is mentioned for fracture, presumably this is per group, but it is not clear if all were analyzed using both methodologies.

25. Figures and legends: most of the figures lack information of the timing that things were measured, for example none of the RNA or western blots have this important information. The osteoclast ones should also indicate whether RANKL was added.

26. Figure 3 legend: please state concentrations of the zinc compounds.

Reviewer #1 (Remarks to the Author):

Major comments

1. I am missing a histological quantification of bone, cartilage and fibrous tissue in the fracture callus sections. This is gold standard for fracture healing studies to evaluate endochondral ossification. It would be reasonable to hypothesis, that zinc might also influence cartilaginous cells.

: Thank you for your comment. We added histological quantification of bone, cartilage, and fibrous tissue in the fracture callus sections (Figure 5e).

2. Osteocalcin staining is not specific for osteocytes in the fracture callus. The authors should rephrase this to osteoblasts.

: As the reviewer suggested, we rephrased osteocytes to osteoblasts (Marked in red)

3. The authors propose an involvement of YAP signaling in the effects of zinc on fracture healing. However, they do not show any data regarding this. They should at least stain YAP activation in the fracture callus.

: Thanks for the comment. The results of immunochemical staining for YAP protein were added in Figure 5h as suggested by the reviewer.

Minor comments:

4. line 57: the reference for "zink deficiency interferes with fracture healing" is missing

: As the reviewer pointed out, we added the corresponding reference in the revised manuscript (Line 59, Ref. 21, marked in red).

5. please provide morde details about how the fracture was created. Open or closed?

: We performed open femur fracture surgery, and the details regarding the procedures were added in the Methods section (Marked in red).

6. please describe pain management in more detail. According to international animal ethics, mice should also receive pain management BEFORE a surgery

: Unfortunately, we did not manage pain before surgery. However, we injected a pain reliever immediately after the operation before the anesthesia worn off. Also, we gave it daily until post-operative day 3. As you have pointed out, we ought to include perioperative pain

management in future studies. We added an explanation of pain management in the Methods section.

7. line 358: what do the authors mean with 5 mM? Should this be millimeter? Did the authors exclude the former intact cortex from their analysis and only measure the fracture callus?

: We are sorry to confuse you because of the typing mistake. We corrected 5 mM to 5 mm in the revised manuscript (Line 410), and we also analyzed the callus volume by subtracting cortical bone volume from total volume. In order for readers to understand, we explained it in the revised version (marked in red).

8. abstract: "There are many kinds of medications and supplements available to treat or support patients with fractures" - I don't agree with this sentence, Vit D and calcium are the only broadly available medications/supplementations during fracture healing so far

: We rephrased the sentence (Line 17-18).

9. abstract: "Therefore, we suggest that drug repositioning of polaprezinc would be helpful for patients with fractures, especially in patients with non-union fractures." The authors did not evaluate the effects of zinc in a non-union model, therefore they should skip the last part of the sentence.

: The sentence has been deleted in the revised version.

Reviewer #2 (Remarks to the Author):

Major comments

1. The introduction and discussion don't cover all the relevant literature, and the citations appear to be quite selective throughout. Specifically, the discussion of role of zinc in bone homeostasis is very limited. A basic pubmed search turns up multiple studies that assess various types of zinc treatment during bone healing and these are not covered at all. The discussion of the role of YAP in bone homeostasis and fracture healing is also very limited. Joel Boerkel's lab has published a number of studies addressing this including a recent one looking at fracture healing which may be appropriate to cover. I suggest reworking the introduction and discussion to address this, and to consider addressing the following questions: Does zinc have a specific role in bone formation, such as getting incorporated into

the matrix, or being a cofactor for collagen or other ECM synthesis? How common is zinc deficiency, and could this be exacerbated following injury?

: Thanks for the comments. Parts of the introduction and discussion have been reworked as suggested by the reviewer. The revised sentences are marked in red. Because of the reviewers' comments, we think that the introduction and discussion part of this manuscript have been greatly improved, compared to the previous version. Thanks again for the comments.

2. Not all the conditions of the editorial checklist are met. There is no data availability statement, or ethics statements for animals or humans. The figures should be redone to show individual data points for most of the graphs.

: As you have pointed out, we added ethics statements in the Methods section (Marked in red). The graphs in the revised version has been updated with the graphs with individual data points.

3. The polaprezinc contains covalently bound zinc. The authors imply it is a source of ionic zinc that they expected to be equivalent to zinc sulfate, but this must require some sort of chemical reaction to release the zinc atom. Where does this reaction occur? Will it happen in vitro in the same way as following oral delivery?

: Thanks for the comments of this reviewer. As the reviewer noted, polaprezinc is a chelated form of zinc and another compound, L-carnosine. A review paper by T. Matsukura and H. Tanaka suggested the chemical reaction takes place to release zinc during intestinal absorption *in vivo* [*Biochemistry (Mosc)*. 2000 Jul;65(7):817-23. PMID: 10951100]. The authors also reported that polaprezinc dissociate to L-carnosine and zinc during intestinal absorption (*Arzneimittelforschung*. 1991 Sep;41(9):965-75. PMID: 1796927). Because polaprezinc is dissolvable in acid and appears to have the ability to protect against mucosal lesions, it can be thought that polaprezinc dissociates to two compounds during intestinal absorption (*Physiol Rev* . 2013 Oct;93(4):1803-45. PMID: 24137022). In *in vitro* experiments with polaprezinc, we dissolved polaprezinc in 0.5mM hydrochloric acid solution. We confirmed that the addition of 0.5 mM HCl to the culture medium had no cytotoxicity. We believe that polaprezinc dissolved in an *in vitro* cell culture environment will be dissociated by 0.5 mM HCL into both compounds, L-carnosine and zinc, similarly to *in vivo* conditions.

4. The osteoclast data should really include treatment with more than one concentration of pZinc (Figure 2), and ideally should have osteoclast numbers per well reported. The data in Figures 3E and 4F in particular have no quantitative measure of osteoclast number which is unacceptable.

: In order to be acceptable for the osteoclast data, the related experiments has been re-performed as indicated by the reviewer. The previous data have been replaced with new data performed using different concentrations of polaprezinc (Figure 2b). Also, the numbers of TRAP-positive multinucleated cells were added in this revised manuscript (Figure 2d, 3f and 4j).

5. Figure 3C and 4E have no quantitative measure of mineralization, mineralized area or dye release should be reported.

: As you have pointed out, we added the quantification of alizarin red S staining (Figure 3b, 4g).

6. The YAP activity data and it's interpretation in Figure 4 is problematic and requires revision. Firstly, the applicability of using Hela cells with exogenous YAP added is limited. Second, the conclusion that p-zinc affects YAP expression cannot be concluded from the reporter data shown. The YAP is expressed from a pcDNA construct so presumably lacks most of its natural promoter, and therefore regulation measured in this system should be at the protein level. It was be more applicable to perform the reporter assay in at least one of the cell types evaluated previously using the naturally occurring YAP expression to confirm if the overall increase in YAP increases its signaling. This figure legend also does not mention transfection with the YAP plasmid.

: Thank you for the valuable comments regarding YAP-related experiments. To address the concerns raised by this reviewer, YAP-related experiments were re-performed using appropriate cells (hBMSCs and RAW264.7). The results were incorporated in Figure 4 (Please see Fig. 4d). In the newly added results, we found that both polaprezinc and zinc sulfate increased the luciferase activity of HOP flash vector. Interestingly, polaprezinc was more effective than zinc in upregulating the transcriptional activity of YAP (Fig. 4d). This is probably because the phosphorylation polaprezinc of serine126, which is known to be involved in cytoplasmic retention of YAP, was increased in the zinc sulfate-treated group,

compared to the polaprezinc-treated group (Fig. 4a). Therefore, we concluded that polaprezinc-mediated YAP activation is more effective in the differentiation of osteoblasts and osteoclasts than zinc sulfate-mediated YAP activation.

7. The siRNA studies in Figure 4 lack an siRNA control – it is possible that YAP knockdown affects differentiation in the absence of zinc which is important for understanding the relevance of this knockdown.

: Thank you for your suggestion. We re-performed YAP knockdown-related experiments, including an additional control group, according to the reviewers' comments. As the reviewer expected, siRNA-mediated knockdown of YAP clearly reduced the osteogenic and osteoclastogenic differentiation potentials of hBMSCs and RAW264.7 cells. Polaprezinc treatment enhanced the osteogenic and osteoclastogenic potentials of the cells, whereas YAP knockdown abrogated the effects of polaprezinc. The new data were added in Fig. 4e-j.

8. In vivo study: The fracture parameters reported should include 'total volume' and the overall callus size may change. The histology looking at cartilage vs bone area should also be quantified and presented. Finally the method for counting TRAP+ or OCN+ cells is not mentioned, and the OCN+ cells are called 'osteocytes' in the figure (H) but not in the legend. Depending on the method of analysis, it should be possible to distinguish osteoblasts from osteocytes with relative confidence, but they should only be defined as osteocytes if they are embedded in the bone.

: As you have pointed out, we added histological quantification of bone, cartilage, and fibrous tissue in the fracture callus sections (Figure 5e). In order to prevent any confusion, we excluded the number of osteocyte data. Also, we added quantification analysis of IHC (Figure 5i).

Minor comments:

9. Abstract: the second sentence is not accurate, there are not really approved pharmacotherapies for fracture healing.

: Thank you for your comment. We revised the sentence (Marked in red).

10. Introduction – a lot of the first paragraph is unnecessary. The third paragraph also

contains excessive explanation of why fractures are a problem/costs involved. The introduction can be reworked as suggested above without changing the length dramatically.

: Thank you for this comment. As the reviewer mentioned in Major Comment 1, we have made major revisions to the Introduction Part. The first paragraph was abbreviated and incorporated into the newly written second paragraph because the purpose of this study is to reposition an existing drug and we wanted to highlight and inform the reader of that. The third paragraph has been moved to the Discussion Part. Changed or newly added parts are marked in red.

11. Line 51-53: Can the authors be more specific about the effects of zinc on osteoblast and osteoclast differentiation?

: Thank you for the comment. We rephrased the sentence as follows:

“Our previous studies also showed that zinc is involved in the calcium-calcineurin-NFATc1 signaling pathway to inhibit osteoclast differentiation in mouse bone marrow-derived monocytes (mBMMs) as well as AMP-PKA-CREB signaling pathway to promote osteoblast differentiation in human bone marrow-derived mesenchymal stem cells (hBMSCs)”

12. Section 3.3: there is a lot of unnecessary discussion for a results section in the first few sentences, including unfounded claims about the mechanism of activity in vitro (line 129-130). The reference used to justify the importance of osteoclast activity in fracture healing cites a knockout model where there are defects in both osteoblast and osteoclast differentiation. There are plenty of more appropriate examples of more osteoclast-selective regimes, including antiresorptive treatments. A factor that promotes osteoclastogenesis is potentially a controversial approach, but notably BMP2 promotes osteoclastogenesis in some settings, so it is not without precedent.

: We appreciate the reviewer for this valuable comment. As the reviewer pointed out, there is a lot of unnecessary discussion in the results section. The sentences have been deleted or moved to a Discussion section. The related contents and references were revised or incorporated in the Discussion session.

13. Results: fracture studies – please state the dosage of polaprezinc and justify the choice of this dose.

: As suggested, we described the relevant reference and dosage of polaprezinc we used.

14. Fracture study lines 207 and 209 – the data shown does not indicate changes in bone remodelling. Remodelling is a dynamic process and either a time course or histomorphometric measures are required to make this conclusion.

: Thank you for your comment. We corrected it (the corrected sentences are marked in red).

15. Methods: hBMSC protocol should be covered briefly, at least to include ethical permissions, passage number, and how many donors were used.

: We added ethics statements, passage numbers, and information on donors in the Methods section (marked in red).

16. Methods: 1M HCl is very acidic for a vehicle, please clarify the final concentration following dilution to reassure the readers that it will not affect the final pH.

: I agree with your opinion. When we measured the cytotoxicity of polaprezinc, the vehicle group was treated same concentration of 100 μ M polaprezinc. However, there was no cytotoxicity even through long-term cultivation. Therefore, we thought that the final concentration of 0.5 mM HCl had no cytotoxicity. We explained this in the revised manuscript (marked in red).

17. Methods: ‘10 ng/mL mM of CSF’ in line 287 and 290 looks like a typo

: We are sorry for this confusion due to a typing mistake. We corrected typological errors in the revised manuscript (marked in red).

18. Methods line 301 – it is unclear what this statement about TRAP+ cells means, especially as osteoclast counts are not actually presented in the manuscript

: As you have pointed out, we defined TRAP-positive cells that have more than five nuclei as osteoclasts. We described it in the Methods section (marked in red). Also, the numbers of TRAP-positive multinucleated cells were added in the revised manuscript (Figure 2d, 3f and 4j).

19. Zinquin protocol lines 323-327: was there a washing step in this protocol?

: Thanks for your comment. We did have a washing step, and a detailed explanation of the protocol was added in the Methods section (marked in red)

20. Luciferase assay: a plasmid that expresses Renilla luciferase is not mentioned but is used as a control.

: As you mentioned, we added the information of Renilla plasmid in the Methods section.

21. Fracture model: please describe more thoroughly how the fracture is generated, and specify the method of oral delivery.

: We performed open femur fracture surgery, and details on the procedure you mentioned were added in the Methods section (marked in red).

22. MicroCT methods: Please double check the description from line 358-360. How was the centre of the fracture identified, and how was the original cortical bone excluded from analysis?

: The center of the fracture line was determined between the intact cortical of the fracture using SkyScan CT-Analyzer software according to a reference (*Bone Reports*, Volume 12, 2020, Article 100250).

According to the paper mentioned above, we reanalyzed callus volume and exchanged μ CT quantification data. As you have pointed out, we rewrote the description in the revised manuscript (marked in red).

23. Methods: please double check the use of mM vs mm.

: We are sorry for this typo. We corrected typographical errors in the revised manuscript.

24. In vivo study – it is not clear if the same bones or different ones are used for microCT and

histology. N=10 is mentioned for fracture, presumably this is per group, but it is not clear if all were analyzed using both methodologies.

: Thank you for your comments. We used the same bone samples for micro CT and immunohistochemical analysis. We described this in the revised manuscript (marked in red). In addition, we performed animal experiments in a total of 20 mice, but excluded four mice on the day after operation due to pin displacement or wound dehiscence. Thus, eight mice per group were used for analysis, and the min and max values were excluded in the data we presented. Therefore, we corrected figure legend to n=6, which is we actually used for analysis. We are sorry for the confusion. We corrected and explained it clearly (marked in red).

25. Figures and legends: most of the figures lack information of the timing that things were measured, for example none of the RNA or western blots have this important information. The osteoclast ones should also indicate whether RANKL was added.

: As you have pointed out, we added detailed information on the data in the revised manuscript.

26. Figure 3 legend: please state concentrations of the zinc compounds.

: We added the concentrations of the zinc compounds in the revised manuscript.

REVIEWERS' COMMENTS:

Reviewer #1 (Remarks to the Author):

The authors answered all of my previous questions and added interesting new data to the manuscript. The manuscript improved and is now acceptable.

Reviewer #2 (Remarks to the Author):

The authors have made extensive revisions to the manuscript including addition of substantial new data, and have thoroughly addressed the majority of my comments.

I have two further comments:

#3 – the authors provide a nice explanation of how zinc is released from polaprezinc in their response, could some of this information be added to the manuscript?

#24 – It would be appropriate to mention the exclusion of animals due to the complications mentioned in the methods, this is fine. The exclusion of the min and max values appears problematic, especially if this means excluding different animals for different measurements. If the animals meet defined biological or statistical exclusion criteria, they can be excluded, otherwise the full data should be presented. Fracture studies often show substantial variability for a variety of technical reasons, this is well accepted.

COMMSBIO-21-0946B

REVIEWERS' COMMENTS:

Reviewer #1 (Remarks to the Author):

The authors answered all of my previous questions and added interesting new data to the manuscript. The manuscript improved and is now acceptable.

: We are very grateful for the positive response from Reviewer #1. Based on this reviewer's opinion, our manuscript has been finished in a more structured way. Thank you again.

Reviewer #2 (Remarks to the Author):

The authors have made extensive revisions to the manuscript including addition of substantial new data, and have thoroughly addressed the majority of my comments.

I have two further comments:

#3 – the authors provide a nice explanation of how zinc is released from polaprezinc in their response, could some of this information be added to the manuscript?

: Thanks for the positive reply. The information has been added to the introduction part of the revised manuscript at the suggestion of the reviewer. In order to appeal to the readers of “*Communications Biology*” that the *in vitro* experiments performed in this study using polaprezinc can provide a similar environment for studying the reactions occurring *in vivo* after oral administration, it was considered appropriate to insert this part in the introduction part. The added part is marked in red.

#24 – It would be appropriate to mention the exclusion of animals due to the complications mentioned in the methods, this is fine. The exclusion of the min and max values appears problematic, especially if this means excluding different animals for different measurements. If the animals meet defined biological or statistical exclusion criteria, they can be excluded, otherwise the full data should be presented. Fracture studies often show substantial variability for a variety of technical reasons, this is well accepted.

: Thanks for this valuable reviewer's comment. Our histological data were not analyzed using different animals for different measurements. That is, our histology data were exactly analyzed using the same experimental animals, and the animals excluded from all histological analyzes were identical for all histological graphs. Therefore, as noted by the reviewers, the data meet the defined biological or statistical exclusion criteria. In order to make this part more clearly, we added the related explanation of exclusion criteria to the method section

(marked in red). In addition, raw data from histological analysis was added as a supplementary table, including exclusion data. Thank you again.